

# Incidence and risk factors of hepatitis E virus infection in women with gynecological tumors in Eastern China

Wenye Bai[1,*], Xiao Wu[2,*], Shuchao Zhao[3], Yang Yu[4], Zhongjun Wang[4], Xiu Li[5] and Na Zhou[6]

[1] Department of Hepatobiliary and Pancreatic Surgery, The Affiliated Hospital of Qingdao University, Qingdao, China
[2] Department of Clinical Laboratory, Qingdao Women and Children's Hospital, Qingdao, China
[3] Department of Pathology, The Affiliated Hospital of Qingdao University, Qingdao, China
[4] Department of Clinical Laboratory, The Affiliated Hospital of Qingdao University, Qingdao, China
[5] Department of Obstetrics and Gynecology, Qingdao Municipal Hospital, Qingdao, China
[6] Department of Surgery, The Affiliated Hospital of Qingdao University, Qingdao, China
* These authors contributed equally to this work.

Corresponding authors
Xiu Li, 641634358@qq.com
Na Zhou, zhou_na_love@126.com

## ABSTRACT

**Background:** Recently, there has been increasing interest in the exploration of the association between the hepatitis E virus (*HEV*) infection and malignancies; however, epidemiological data for *HEV* infection among women with a gynecological tumors (GT) are limited. Herein, we investigated the correlation between *HEV* and GT in Chinese women.

**Methods:** We recruited 452 women diagnosed with a primary GT and 452 healthy volunteers to investigate the possible routes and risk factors for *HEV* infection. The serum antibody levels of anti-*HEV* IgG and IgM were measured by enzyme-linked immunoassays once a year.

**Results:** After a median follow-up time of 5.4 years (range 4 to 7 years), the overall detection rate of anti-*HEV* antibodies in patients with GT and in controls were 69/452 (15.27%) and 23/452 (5.09%) (*P* = 0.001), respectively. The seroprevalence of anti-*HEV* IgG antibodies was significant higher in patients with GT (15.27%) than in healthy controls (5.09%) (*P* = 0.001). Moreover, 13 (2.88%) patients with GT were positive for IgM antibodies, while only 4 (0.88%) healthy controls tested positive for anti-*HEV* IgM antibodies (*P* = 0.028). The highest prevalence of *HEV* antibodies were detected in patients with ovarian borderline tumors (40%), followed by patients with ovarian cancer (20.54%) and endometrial cancer (18.46%). Multivariable analysis revealed that contact with dogs (OR, 1.88; 95% CI [1.10–3.22]; *P* = 0.015) and a history of anti-tumor chemotherapy (OR, 1.85; 95% CI [1.07–3.20]; *P* = 0.028) were independent risk factors for *HEV* infection.

**Conclusion:** Overall, the present study showed that patients with GT are more susceptible to *HEV* infection in Eastern China, particularly in patients with ovarian borderline tumors. Thus, effective strategies are needed to reduce *HEV* infection in patients with GT.

## INTRODUCTION

The hepatitis E virus (*HEV*) is a single-stranded RNA virus, which is estimated to have infected nearly 20 million individuals worldwide (*Ma et al., 2022*). *HEV* has been classified into four major genotypes (*HEV*1-4) and 24 sub-types (*Aslan & Balaban, 2020*). *HEV* genotypes 3 and 4 can be transmitted from animals to humans *via* the fecal–oral route (*Busara et al., 2024*). *HEV* infection is usually a self-limiting disease. Sometimes they are completely non-specific symptoms, but often there are liver symptoms as well (*Hoofnagle, Nelson & Purcell, 2012*; *Kamar et al., 2012*). However, in immune-deficient patients, including patients with tumors or autoimmune diseases, *HEV* infection may cause liver failure and death (*Elfert et al., 2018*; *Webb & Dalton, 2020*).

A high incidence of *HEV* infection has been found in patients with cancer (*Bai et al., 2018*; *Lin et al., 2023*). In one study, *Bai et al. (2018)* demonstrated that nearly 26% of patients with cancer were seropositive for anti-*HEV* antibodies, which indicates either past or current *HEV* infection. This seroprevalence is considerably higher than the 13% positivity rate observed in the control group, a statistically significant difference suggesting that cancer patients may be at increased risk of *HEV* infection (*Bai et al., 2018*). Another study conducted by *Chiu et al. (2022)* reported a latent relationship between *HEV* and hematologic malignancies. In addition, *Lin et al. (2023)* analyzed the relationship between *HEV* infection and the risk for 17 types of cancer, finding a significant association between *HEV* infection and gastric cancer. Together, this evidence indicates that *HEV* infection may be a significant risk factor for malignancy.

Gynecological tumors (GT) include cancers that develop in the female reproductive system. According to the position of the cancer, GT can be classified as either external genital tumors, vaginal tumors, uterine tumors, ovarian tumors, or fallopian tube tumors, *etc*. GTs may further be classified as benign, malignant, or borderline. It is generally believed that the development of GT can be driven by genetic factors, pathogen infection, and physical and chemical factors. For example, human papillomavirus (*HPV*) is associated with tumor progression in *HPV*-associated cervical carcinoma (*Senapati, Senapati & Dwibedi, 2016*). *Bai et al. (2018)* also showed that patients with ovarian cancer were more to susceptible *HEV* infection, suggesting a potential association between *HEV* infection and ovarian tumors development.

There are many risk factors associated with *HEV* infection, including age, region of residence, and contact with infected animals, among other. Several epidemiological investigations of *HEV* infection in patients with malignancies have been conducted in recent years (*Bai et al., 2018*; *Chiu et al., 2022*). Further, other studies have shown that receiving blood transfusion and anti-tumor chemotherapy can also increase the risk of *HEV* infection (*Bettinger et al., 2018*; *Boutrouille et al., 2007*; *Donald & Peter, 2018*; *Okumura et al., 2023*). However, data regarding *HEV* infection in patients with benign GT is scarce, and the prevalence and potential risk factors for this virus in such patients are currently unknown. Thus, the aim of this study was to explore the risk of *HEV* infection in women with GT, and to clarify the potential risk factors for this patient group.

## METHODS

### Ethics statement

This project was approved by the Ethics Committee of the Affiliated Hospital of Qingdao University (QYFY WZLL 28350). All participants provided written informed consent to participate.

### Study cohort and sociodemographic data

Between January 2016 and December 2019, 1,029 volunteers, including 543 women diagnosed with a primary GT and 486 healthy controls, were recruited to participate this study. Patients with GT ranged in age from 21–69 years old. Healthy controls were randomly invited from among women who participated in health screenings at the Affiliated Hospital of Qingdao University. Healthy controls are not diagnosed with any gynecological disorders when they were recruited. Participants who tested positive for anti-*HEV* antibodies or were treated with intravenous immunoglobulin (Ig) before blood collection were excluded. All volunteers were followed up until December 2023, and data regarding behavioral characteristics and patient survival were collected. The tests will be terminated when the volunteers infect with *HEV*, and the questionnaire will be given. The questionnaire of those negative for *HEV* antibodies was given on December 2023. Sociodemographic and lifestyle behavioral data were collected from participants using a structured questionnaire, as described by *Wang et al. (2022)*. Clinical disease data (including tumor type, serum markers) were collected from medical records supplemented by the patients.

### Sample collection and serological assay

Venous blood samples of ~5 mL were collected from volunteers once a year. After collection, blood samples were centrifuged at 3,000 rpm for 10 min at room temperature to collect serum. Serum samples were collected and stored at −80 °C until examination to ensure the integrity and reliability of the results (*Wang et al., 2022*). The ELISAs were completed within 3 months after blood collection.

   Enzyme-linked immunosorbent assay (ELISA) kits (Wantai Bio, Beijing, China) were used to test for anti-*HEV* IgG and IgM antibodies. The sensitivity and specific of the *ELISA* are 98.5% and 99.1%, in accordance with the manufacturer's instructions. Briefly, 100 μl sample diluent was pipetted into a single well of a 96-well plate, and supplemented with 10 μl serum. After incubation for 30 min at 37 °C, the well plates were washed five times. Subsequently, 100 μl of horseradish peroxidase-conjugated enzyme labeled HEV-Ag was added to each well, and incubated in the dark allowed for 30 min at 37 °C. After washing, chromogenic solution A (50 μL) and chromogenic solution B (50 μL) were added to the 96-well plate and incubated for 15 min. Termination solution (50 μL) was then added into the well to stop the reaction. The optical density (OD) values were measured at 450 nm using Labsystems Multiskan RC micro-plate reader. Positive and negative control sera were included in each plate. The cutoff value was calculated as the mean of negative controls plus 0.26. Results equal to or greater than the cutoff value were considered as positive.

## Statistical analyses

All statistical analyses were performed using SPSS 22.0 (IBM, Armonk, NY, USA). The association between the anti-*HEV* antibody positive rate and socio-demographic and clinical data were analyzed by chi-square test or Fisher's exact test. Data associated with *HEV* infection in univariate analysis ($P \leq 0.2$) were included in a multivariate logistic regression analysis to define independent risk factors of *HEV* infection. The adjusted odds ratio (OR) and 95% confidence interval (CI) were calculated using logistic regression analysis. Results with a *P*-value of < 0.05 were considered significant.

# RESULTS

## Epidemiological profile and risk factors for patients with GT and *HEV* infection

After a median follow-up time of 5.4 years (range 4 to 7 years), 452 women diagnosed with a primary GT and 452 healthy controls obtained complete follow-up data. The anti-*HEV* antibody presence was tested in these 904 participants (452 patients with GT and 452 controls). The overall incident rate of *HEV* infection in patients with GT and in controls was 69/452 (15.27%) and 23/452 (5.09%) ($P = 0.001$), respectively, representing a significantly higher level in GT patients ($P = 0.001$). In addition, 13 (2.88%) patients with GT were positive for IgM antibodies, while only 4 (0.88%) healthy controls were anti-*HEV* IgM antibody positive ($P = 0.028$) (Table 1). Univariate analysis showed that patients' age, contact with dogs, source of drinking tap water, and history of anti-tumor chemotherapy were all associated with *HEV* seroprevalence in patients with GT. The detailed data are shown in Table 2. All socio-demographic and clinical treatment variables with $P \leq 0.2$ on analysis (age, contact with dogs, contact with pigs, source of drinking water, and history of anti-tumor chemotherapy) were included in the subsequent multivariate analysis. This analysis revealed that contact with dogs (OR, 1.88; 95% CI [1.10–3.22]; $P = 0.015$) and a history of anti-tumor chemotherapy (OR, 1.85; 95% CI [1.07–3.20]; $P = 0.028$) were independent risk factors for *HEV* infection in patients with GT (Table 3). In additional, we tested the serum Glutamic Oxaloacetic transaminase (SGOT) and serum glutamic pyruvic transaminase (SGPT) values of volunteers who presented positive anti-*HEV* antibody. We found there was no difference in SGOT and SGPT values between each group of patients and the control group (Table 4).

## *HEV* antibody prevalence in patients with different GT histological types

The levels of *HEV* exposure in patients with different GT histological types are presented in Table 5. The highest prevalence of *HEV* antibodies were detected in patients with ovarian borderline tumor (40%), followed by patients with ovarian cancer (20.54%) and endometrial cancer (18.46%) ($P < 0.05$). Overall, 37 cancer patients died during the study period, none of them infected whit *HEV*. In addition, among patients with GTs, nearly 80% of *HEV* infection cases were acquired within 3 years of diagnosis, while in healthy controls, the *HEV* infection rate did not present any obvious temporal characteristics (Fig. 1).

**Table 1 Combined *HEV* IgG and IgM antibodies positive rate in patients with gynecological tumor and healthy controls.**

| Sero-reaction | Patients with a GT (n = 452) | | Healthy controls (n = 452) | | Patients with a GT *vs.* Healthy controls |
|---|---|---|---|---|---|
| | No. positive | % | No. positive | % | *P*[a] |
| IgG | 69 | 15.27 | 23 | 5.09 | 0.001 |
| IgM | 13 | 2.88 | 4 | 0.88 | 0.028 |
| IgG⁺/IgM⁺ | 13 | 2.88 | 4 | 0.88 | 0.028 |
| IgG⁺/IgM⁻ | 56 | 12.39 | 22 | 4.87 | 0.001 |
| IgG⁻/IgM⁺ | 0 | 0 | 0 | 0 | 1 |
| Total | 69 | 15.27 | 23 | 5.09 | 0.001 |

Note:
[a] Chi-square test or Fisher's exact test.

**Table 2 Incidence of *HEV* infection in patients with gynecological tumor and health controls in Eastern China.**

| Characteristic | Patients with a gynecological tumor (n = 452) | | | | Healthy controls (n = 452) | | | |
|---|---|---|---|---|---|---|---|---|
| | Prevalence of *HEV* infection | | | | Prevalence of *HEV* infection | | | |
| | No. tested | No. positive | % | *P*[a] | No. tested | No. positive | % | *P*[a] |
| Age (years) | | | | | | | | |
| ≤30 | 95 | 10 | 10.53% | 0.008 | 65 | 6 | 9.23% | 0.267 |
| 31–50 | 109 | 9 | 8.26% | | 119 | 8 | 6.72% | |
| 50–70 | 188 | 35 | 18.62% | | 233 | 12 | 5.15% | |
| >71 | 60 | 15 | 25.00% | | 35 | 0 | 0.00% | |
| Residence area | | | | | | | | |
| Urban | 251 | 42 | 16.73% | 0.332 | 240 | 15 | 6.25% | 0.629 |
| Rural | 201 | 27 | 13.43% | | 212 | 11 | 5.19% | |
| Contact with cats | | | | | | | | |
| Yes | 141 | 24 | 17.02% | 0.485 | 118 | 8 | 6.78% | 0.577 |
| No | 311 | 45 | 14.47% | | 334 | 18 | 5.39% | |
| Contact with dogs | | | | | | | | |
| Yes | 209 | 42 | 20.10% | 0.008 | 107 | 13 | 12.15% | 0.001 |
| No | 243 | 27 | 11.11% | | 345 | 13 | 3.77% | |
| Contact with pigs | | | | | | | | |
| Yes | 74 | 15 | 20.27% | 0.191 | 89 | 6 | 6.74% | 0.655 |
| No | 378 | 54 | 14.29% | | 363 | 20 | 5.51% | |
| Consumption of raw/undercooked meat | | | | | | | | |
| Yes | 107 | 19 | 17.76% | 0.412 | 75 | 6 | 8.00% | 0.360 |
| No | 345 | 50 | 14.49% | | 377 | 20 | 5.31% | |
| Consumption of raw vegetables | | | | | | | | |
| Yes | 78 | 10 | 12.82% | 0.509 | 204 | 14 | 6.86% | 0.358 |
| No | 374 | 59 | 15.78% | | 248 | 12 | 4.84% | |
| Exposure to soil | | | | | | | | |
| Yes | 269 | 41 | 15.24% | 0.986 | 144 | 10 | 6.94% | 0.457 |
| No | 183 | 28 | 15.30% | | 308 | 16 | 5.19% | |

(Continued)

| Characteristic | Patients with a gynecological tumor (n = 452) | | | | Healthy controls (n = 452) | | | |
|---|---|---|---|---|---|---|---|---|
| | Prevalence of HEV infection | | | | Prevalence of HEV infection | | | |
| | No. tested | No. positive | % | $P^a$ | No. tested | No. positive | % | $P^a$ |
| Source of drinking water | | | | | | | | |
| Tap | 346 | 62 | 17.92% | 0.005 | 266 | 14 | 5.26% | 0.593 |
| River | 106 | 7 | 6.60% | | 186 | 12 | 6.45% | |
| Occupation | | | | | | | | |
| Farmer | 278 | 44 | 15.83% | 0.675 | 296 | 20 | 6.76% | 0.206 |
| Worker | 174 | 25 | 14.37% | | 156 | 6 | 3.85% | |
| History of abortion | | | | | | | | |
| Yes | 73 | 12 | 16.44% | 0.761 | 117 | 5 | 4.27% | 0.425 |
| No | 379 | 57 | 15.04% | | 335 | 21 | 6.27% | |
| History of chemotherapy | | | | | | | | |
| Yes | 173 | 32 | 18.50% | 0.004 | | | | |
| No | 279 | 37 | 1.46% | | | | | |
| History of blood transfusion | | | | | | | | |
| Yes | 126 | 20 | 8.20% | 0.823 | | | | |
| No | 326 | 49 | 13.31% | | | | | |

**Note:**
[a] Chi-square test.

**Table 3 Multivariable analysis of patients with gynecological tumor and healthy controls and the association of characteristics with HEV infection.**

| Characteristic | | Adjusted odds ratio[a] | 95% CI[b] | $P^c$ |
|---|---|---|---|---|
| Contact with dogs | Yes *vs.* No | 1.88 | [1.10–3.22] | 0.015 |
| Contact with pigs | Yes *vs.* No | 1.62 | [0.84–3.13] | 0.15 |
| Source of drinking water | River *vs.* Tap | 0.44 | [0.19–1.03] | 0.059 |
| History of chemotherapy | Yes *vs.* No | 1.85 | [1.07–3.20] | 0.028 |

**Notes:**
[a] Adjusted by age.
[b] Confidence interval.
[c] Multivariate logistic regression analysis.

## DISCUSSION

Based on reports by the World Health Organization, viral hepatitis is responsible for approximately 1.45 million deaths globally each year (*World Health Organization, 2016*; *Tjan, 2016*). HEV infection has been shown to cause liver damage, accounting for ~3.3% of all viral hepatitis mortalities (*Primadharsini, Nagashima & Okamoto, 2019*). Thus, HEV infection is now recognized as a significant rising global burden. The HEV seroprevalence in patients with cancer has been increasingly explored in recent years including hepatocellular carcinoma (HCC) (*Shen et al., 2023*; *Xu et al., 2017*; *Yin & Kan, 2023*), gastric cancer, (*Chiu et al., 2022*; *Webb & Dalton, 2020*), and lung cancer

**Table 4 The correction between *HEV* infection and the serum glutamic oxaloacetic transaminase (SGOT) and serum glutamic pyruvic transaminase.**

| Clinical diagnosis | No. tested | High SGOT | % | $P^a$ | High SGPT | % | $P^b$ |
|---|---|---|---|---|---|---|---|
| Gynecological tumor | 69 | 13 | 18.84 | 0.87 | 17 | 24.64 | 0.24 |
| Ovarian borderline tumor | 6 | 2 | 33.33 | 0.39 | 1 | 16.67 | 0.27 |
| Ovarian cancer | 23 | 5 | 21.74 | 0.71 | 7 | 30.43 | 0.35 |
| Endometrial cancer | 12 | 1 | 8.33 | 0.48 | 3 | 25 | 0.33 |
| Cervical squamous cell carcinoma | 15 | 3 | 20 | 0.84 | 3 | 20 | 0.44 |
| Ovarian mucinous cystadenoma | 3 | 1 | 33.33 | 0.51 | 2 | 66.67 | 0.085 |
| Uterine leiomyoma | 3 | 0 | 0 | 0.59 | 0 | 0 | 0.68 |
| Ovarian cystic mature teratoma | 7 | 1 | 14.229 | 0.62 | 1 | 14.29 | 0.68 |

Note:
As compared with 17.39% (4/23) higher SGOT[a] and 13.04% (3/23) higher SGPT[b] in controls, respectively.

**Table 5 The correction between clinical pathology diagnosis and incidence of *HEV* in patients with gynecological tumor.**

| Clinical diagnosis | No. tested | No. positive | % | $P^a$ |
|---|---|---|---|---|
| Gynecological tumor | 452 | 69 | 15.27% | 0.001 |
| Ovarian borderline tumor | 15 | 6 | 40% | 0.001[*] |
| Ovarian cancer | 112 | 23 | 20.54% | 0.001 |
| Endometrial cancer | 65 | 12 | 18.46% | 0.001 |
| Cervical squamous cell carcinoma | 84 | 15 | 17.86% | 0.001 |
| Ovarian mucinous cystadenoma | 26 | 3 | 11.54% | 0.16[*] |
| Uterine leiomyoma | 41 | 3 | 7.32% | 0.47[*] |
| Ovarian cystic mature teratoma | 109 | 7 | 6.42% | 0.58 |

Notes:
[a] As compared with 5.09% seroprevalence of anti-*HEV* antibodies in controls (23/452).
[*] Fisher's exact test were used.

(*Okumura et al., 2023*). In addition, in our previous study, we found a significantly higher seroprevalence of anti-*HEV* antibodies in patients with ovarian cancer than in controls (*Bai et al., 2018*). However, whether tumors promote *HEV* infection or potential routes for *HEV* infection in these patients group remains unclear. Thus, we conducted the present study to assess this situation. Our results showed that cancer could increase the risk of infection by *HEV*.

After 6 years of monitoring, we found a significantly higher detection rate of anti-*HEV* IgG antibodies in patients with GT (15.27%, 69/452) than in healthy controls (5.09%, 23/452) at the end of the follow-up period. These data suggested that patients with GT are more susceptible to *HEV* infection. Moreover, the seroprevalence of *HEV* in patients with malignancies was higher than that in patients with benign tumors, with particularly high rates observed in patients with ovarian and endometrial cancer. Patients with malignant tumors commonly show immune deficiencies, resulting in an inability to form effective responses against *HEV* (*Lenglart et al., 2023*; *Lin et al., 2023*; *Yin & Kan, 2023*). Moreover, anti-*HEV* antibodies were most commonly detected within 3 years of GT diagnosis, while the *HEV* infection rate in healthy women did not present any obvious temporal

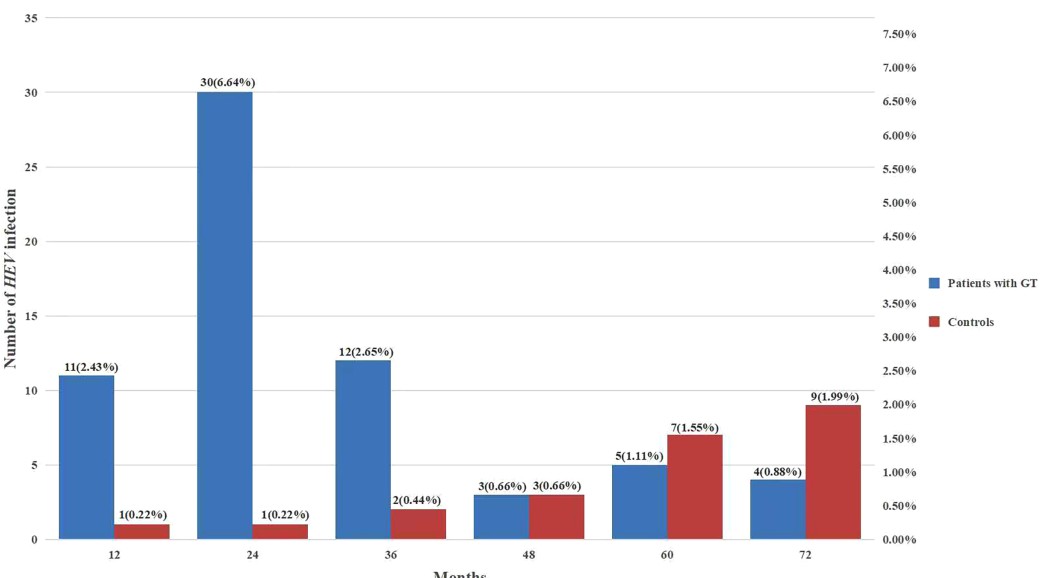

**Figure 1 Comparison between *HEV* serostatus and follow-up time.**

characteristics. Another interesting result of this study is our finding that patients with ovarian borderline tumors presented with the highest incidence rate of *HEV*, which suggested patients with ovarian borderline tumors were the most sensitive to *HEV* infection.

Previous studies have indicated that the *HEV* infection rate in healthy individuals may increase with age, due to an overall lifetime exposure to *HEV* among older people (*Ouyang et al., 2024*). However, one study showed that in cancer patients, *HEV* seroprevalence was significantly higher in young patients (*Lin et al., 2023*). Some researchers have speculated that the change in immune function and hormonal levels caused by aging may account for this discrepancy. In our study, we found that patients with GTs older than 70 years had the highest incidence for *HEV* infection, while in healthy controls, younger participants had a greater risk of *HEV* exposure. The possible reason for this phenomenon is that women with GTs are immunodeficient, and older patients maybe have more fragile immune system to against *HEV* infections. Overall, this study revealed a potential correlation between *HEV* infection and age in GTs patients; however, further studies are needed to confirm the potential mechanisms.

The fecal–oral route is an important mode of transmission for *HEV*. Contaminated drinking water, contact with pigs and cats, and exposure to feces are the most common risk factors for *HEV* infection (*Michelle et al., 2013*). In immunocompromised patients, *HEV* can also be transmitted from blood products (*Hoofnagle, Nelson & Purcell, 2012*). *Kogias et al. (2023)* demonstrated that in hemodialysis patients, *HEV* infection was significantly associated with area of residence and contact with pork. However, the risk factors for *HEV* infection among patients with malignant tumors have not been well demonstrated. In our study, multivariate analysis showed that contact with dogs was an independent risk factor for *HEV* infection in women with GTs and healthy controls. This result is consistent with a

study conducted by *Bai et al. (2018)*, in which cancer patients in contact with dogs at home harbored the highest *HEV* seroprevalence. In addition, one survey including nearly 4,500 dogs in Southwestern China identified anti-*HEV* antibodies in 36.55% of stray city dogs. This data suggests that a high *HEV* seroprevalence in dogs and humans exposed from dogs should be considered an urgent public health concern (*Zeng et al., 2017*). Although patients with GTs can be infected with *HEV* through contact with dogs, little attention has been paid to this phenomenon. Therefore, it will be necessary to further investigate this risk factor for *HEV* infection in cancer patients, particularly those with GTs, in order to reduce the transmission of *HEV*.

Chemotherapy was identified as another risk factor for *HEV* infection in patients with GT in our study. This result is in agreement with other studies, which identified acute *HEV* infection in some tumor patients during anti-tumor chemotherapy (*Bettinger et al., 2018*; *Lenglart et al., 2023*). Adjuvant chemotherapy combined with targeted therapy is an important treatment strategy for gynecologic cancer. For example, the combination of paclitaxel and carboplatin is the first-line clinical treatment for gynecologic malignancies. However, this management strategy commonly causes liver injury in patients. Moreover, the combination of paclitaxel and carboplatin adjuvant chemotherapy is usually used for patients within 2 years after diagnosis. Therefore, it is reasonable to propose chemotherapy may increase the risk of *HEV* infection.

This study has some limitations which should be mentioned. First, the sample size was relatively small and therefore is not representative of the entire Chinese population. Second, we did not conduct *HEV* RNA tests to exclude false positives caused by the use of ELISA diagnostic equipment; thus, the influence of false positivity caused by the methodology is uncertain. Third, there might have been cases who did not detect anti-*HEV* antibodies when they had an *HEV* infection due to the difference in time between *HEV* infection and sample collection.

## CONCLUSIONS

The results of the present study show that patients with GTs are more susceptible to *HEV* infection, especially in patients with ovarian borderline tumors. Contact with dogs and treatment with chemotherapy are independent risk factors for this virus infection. Thus, effective strategies are urgently needed to reduce *HEV* infection in patients with GT.

### Funding

The authors received no funding for this work.

### Competing Interests

The authors declare that they have no competing interests.

### Author Contributions

- Wenye Bai conceived and designed the experiments, performed the experiments, authored or reviewed drafts of the article, and approved the final draft.

- Xiao Wu performed the experiments, authored or reviewed drafts of the article, and approved the final draft.
- Shuchao Zhao performed the experiments, authored or reviewed drafts of the article, and approved the final draft.
- Yang Yu performed the experiments, analyzed the data, prepared figures and/or tables, and approved the final draft.
- Zhongjun Wang performed the experiments, prepared figures and/or tables, and approved the final draft.
- Xiu Li performed the experiments, prepared figures and/or tables, and approved the final draft.
- Na Zhou conceived and designed the experiments, analyzed the data, prepared figures and/or tables, authored or reviewed drafts of the article, and approved the final draft.

### Human Ethics
The following information was supplied relating to ethical approvals (*i.e.*, approving body and any reference numbers):

This project was approved by the Ethics Committee of the Affiliated Hospital of Qingdao University.

### Data Availability
The raw data are available in the Supplemental File.

### Supplemental Information
Supplemental information for this article can be found online at http://dx.doi.org/10.7717/peerj.18747#supplemental-information.

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
