# Peer review of "Incidence and risk factors of hepatitis E virus infection in women with gynecological tumors in Eastern China"

_PeerJ, doi:10.7717/peerj.18747_

## Round 0.1 · original submission · Major Revisions

As indicated in the reviews below, while the reviewers believe the research is interesting and important, there are sufficient concerns regarding the manuscript, that major revisions will be needed before any decision can be made regarding acceptance and publication in PeerJ. In particular, there was uncertainty regarding the cause-effect relationships between infection with HEV and the occurrence of gynecological tumors. A revised manuscript should clearly differentiate between direct causal relationships, associative relationships, the direction of the relationships, and the extent to which underlying risk factors may play a role.

Reviewer 1 ·

Basic reporting

In the manuscript titled ‘Seroprevalence and Risk Factors of Hepatitis E Virus Infection in Women with Gynecological Tumors in Eastern China’, Wenye Bai et al. identified risk factors for HEV infection in patients with gynecological tumors and healthy controls. The story is interesting, the essential data has been deposited, the analysis seems correct, and the results support the main claims. Some specific comments are listed as follows:
1. While the manuscript is understandable overall, there are language issues throughout the manuscript that may require a fluent English speaker to fix. For example, in line 31 ‘patients with GT in controls’, in line 70, ‘were more susceptible to HEV’. Please check the whole manuscript and fix similar issues.
2. In line 26, the authors stated that the aim of the study is to investigate the correlation between HEV and GT in Chinese women. It is not correct as most parts of the manuscript are talking about risk factors of HEV infection. To my understanding of the manuscript, the risk factors like age and contact with dogs are validated in both GT patients and controls.
3. In lines 41-42, the authors claimed that HEV infection is a risk factor for GT. This is clearly not supported by the manuscript. The whole study is identifying risk factors for HEV infection instead of identifying HEV infection as a risk factor.
4. In lines 50-51, you will need a reference to support claims like ‘HEV genotypes 1 and 2 can be transmitted from animals to humans via the fecal-oral route’. Please check the whole manuscript to fix similar issues.
5. In lines 80-81, I do not fully understand the sentence ‘to clarify the potential susceptibility factors for this patient group’. Please rephrase.

Experimental design

1. In lines 89-90, were there real ‘healthy controls’ or are they just not diagnosed with GT but with other gynecological disorders? Please clarify.

Validity of the findings

1. In the discussion section, the authors claim the linkage between HEV infection and malignant tumor/malignancy (lines 161-162, 167-169). However, no clinical information of malignancy or tumor-staging is included in the study. Were the 452 patients with GT all malignant? To support a linkage between HEV infection and malignancy, at least both benign and malignant tumors should be included and compared in the study.
2. In lines 180-182, the authors reasoned that ‘HEV may be involved in the malignant transformation of ovarian tumors’ based on the results that ‘patients with ovarian borderline tumors presented with the highest seroprevalence of HEV’. First of all, ‘ovarian borderline tumors’, although not completely benign, are still debatable to be considered as malignant. Secondly, even if ‘ovarian borderline tumors’ are malignant, this result can only suggest that malignant patients are more susceptible to HEV infection instead of HEV being involved in or causing malignant transformation as claimed by the authors. Here, the results suggest that HEV infection may be the consequence but not the cause of malignancy.
3. Similar logical mistakes as mentioned above are common in the discussion section. I cannot point out each of them, but please go through the discussion section carefully to fix.

Reviewer 2 ·

Basic reporting

a. The manuscript is well-written and clear, with almost no errors. The only error I identified was a missing word in Line 196 – “while older women may have more fragile immune system to against HEV infections.”

b. Can the authors clarify “negative control plus 0.26” in Line 114. Was 0.26 added to correct readings from the ELISA plate reader?

c. The authors have also provided sufficient to understand their research.

d. The provided tables are labelled and relevant. Raw data is provided and accessible. However, the consent forms and ethics statements are not in English.

Experimental design

The authors have adequately described the need for their research, which addresses the gap in understanding overlapping subsets of HEV patients and patients with gynecological tumors. To this end, they analyzed retrospective blood samples from healthy controls and patients with gynecological tumors for anti-HEV antibodies using ELISA.

Anti-HEV IgM usually last for up to 6 months, while Anti-HEV IgG has been known to last for years. Is it possible that there might have been cases where the authors did not find anti-HEV antibodies (particularly IgM) in patients who had an HEV infection, possibly because of the difference in time between disease and sample collection?

Validity of the findings

In line 181, the authors suggest, “This result suggests that HEV may be involved in the malignant transformation of ovarian tumors”. However, due to the associative nature of this study, such a conclusion cannot be drawn.

The findings of this research are statistically significant. The authors might consider including the following points in their discussion to add to their findings.

a. If the authors have access to clinical data from the HEV patients, can they comment on whether the disease exhibited more morbidity (e.g., higher SGOT, SGPT values) in one group (e.g., GT HEV cases) compared to the other?

b. Are medications administered for ovarian borderline tumor particularly hepatotoxic, or known to suppress immunity?

c. 17% of HEV+ GT patients used tap water, compared to 5.26% in HEV+ control individuals. Can the authors comment on any socioeconomic differences between the two groups based on this or other observations?

Reviewer 3 ·

Basic reporting

The directionality of the research question is very muddled throughout the paper. In the introduction and discussion/conclusion, the researchers talk both about how hepatitis E could potentially lead to an increased risk of tumors AND how the tumors (or their treatment) could lead to an increased risk of hepatitis E. Which direction was the original research question that inspired this project? The design of this study can only test one direction: What is the risk of hepatitis E in patients with gynecological tumors? This needs to be made clear in the introduction and discussion.
Examples: Line 61-62, 69-71, 79-81, 163 to 163 (importantly this statement suggests this study could determine the direction of causality between gt and HEV, this is not the case).

It seems like the is study measures incidence (negative to positive over a time period), rather than prevalence (seroprevalence at one point in time) given the longitudinal nature of the follow up. The authors should justify why they call this prevalence rather than incidence).

Line 50 should read “genotypes 3 and 4”.

Line 52: Sometimes they are completely non-specific symptoms, but often there are liver symptoms as well

Line 57: Many studies have found the seroprevalence in the general population to be about 25%. What was the seroprevalence in the comparison population in this specific study?

The paragraph from lines 141-148 seems out of place. It would make more sense to move this next to the other seroprevalence section at the beginning of the results.

Include the seroprevalence of HEV in the patients that died during the study period.

The word “this” in line 156 implies the current study, better word choice is needed.

The type of test to give the p-values should be indicated in the table as well as the text.

Experimental design

More information about the study design is needed. The methods state that blood was drawn every year, but there are no results suggesting this. Which blood sample was used to get the seroprevalence estimates? How often was the questionnaire given? Given that these are time varying exposures, which blood sample and/or questionnaire was used for this analysis? If the questionnaire was only given at baseline, this needs to be listed as a major limitation given the time varying exposures and the HEV infection could have occurred up to 7 years after the questionnaire.

More details about the logistic regression need to be included. Table 2 clearly has the cases and controls analyzed separately. Are the controls not included in the analysis in Table 3 at all? Or are they included? What is the purpose of the healthy controls if they are not included in the logistic regression?

Where there any disruptions in the planned follow up due to COVID shut downs?

Some information about the sensitivity and specific of the ELISA test should be given. How long after blood collection were the ELISAs completed.

How can the minimum follow up time be 5 years given the recruitment went until August 2021 and follow up ended in December 2023?

Provide confidence intervals around the seroprevalence estimates.

Validity of the findings

More information about the study population is needed. How many patients were screened but not included? What was the refusal rate? If the control participants were matched, how could they be randomly invited participate?

The data to support the statement in lines 177-178 is not presented in the results.

The assertion in lines 181-182 is too strong for the results of this study to support and needs to be toned down.

188-189: This could also be explained by overall lifetime exposure to HEV, not greater transmission.

Lines 193-198 seem to be contradictory to what you found. Please revise this section.

Lines 225-229: More discussion of the limitations should be included. Self-reported data, time varying exposures, etc.

Additional comments

Overall, this is an interesting study on the risk for hepatitis E in an understudied population. However, there are some issues that need to be addressed prior to acceptance.

---

## Round 0.2 · accepted · Accept

Thank-you for the revised version of your manuscript. We are pleased to accept it for publication.

Reviewer 1 ·

Basic reporting

All my concerns have been addressed.

Experimental design

All my concerns have been addressed.

Validity of the findings

All my concerns have been addressed.